# Phosphomimetic S207D Lysyl–tRNA Synthetase Binds HIV-1 5′UTR in an Open Conformation and Increases RNA Dynamics

**DOI:** 10.3390/v14071556

**Published:** 2022-07-16

**Authors:** William A. Cantara, Chathuri Pathirage, Joshua Hatterschide, Erik D. Olson, Karin Musier-Forsyth

**Affiliations:** 1Department of Chemistry and Biochemistry, The Ohio State University, Columbus, OH 43210, USA; pathirage.1@buckeyemail.osu.edu (C.P.); jhatt@pennmedicine.upenn.edu (J.H.); erikolson9@gmail.com (E.D.O.); 2Center for Retrovirus Research, The Ohio State University, Columbus, OH 43210, USA; 3Center for RNA Biology, The Ohio State University, Columbus, OH 43210, USA; 4Ohio State Biochemistry Program, The Ohio State University, Columbus, OH 43210, USA

**Keywords:** human immunodeficiency virus type 1, lysyl–tRNA synthetase, 5′ untranslated region, selective 2′-hydroxyl acylation analyzed by primer extension, RNA structure, RNA dynamics, viral RNA, tRNA^Lys3^ primer, tRNA-like element, small-angle X-ray scattering

## Abstract

Interactions between lysyl–tRNA synthetase (LysRS) and HIV-1 Gag facilitate selective packaging of the HIV-1 reverse transcription primer, tRNA^Lys3^. During HIV-1 infection, LysRS is phosphorylated at S207, released from a multi-aminoacyl–tRNA synthetase complex and packaged into progeny virions. LysRS is critical for proper targeting of tRNA^Lys3^ to the primer-binding site (PBS) by specifically binding a PBS-adjacent tRNA-like element (TLE), which promotes release of the tRNA proximal to the PBS. However, whether LysRS phosphorylation plays a role in this process remains unknown. Here, we used a combination of binding assays, RNA chemical probing, and small-angle X-ray scattering to show that both wild-type (WT) and a phosphomimetic S207D LysRS mutant bind similarly to the HIV-1 genomic RNA (gRNA) 5′UTR via direct interactions with the TLE and stem loop 1 (SL1) and have a modest preference for binding dimeric gRNA. Unlike WT, S207D LysRS bound in an open conformation and increased the dynamics of both the PBS region and SL1. A new working model is proposed wherein a dimeric phosphorylated LysRS/tRNA complex binds to a gRNA dimer to facilitate tRNA primer release and placement onto the PBS. Future anti-viral strategies that prevent this host factor-gRNA interaction are envisioned.

## 1. Introduction

The fundamental property that distinguishes retroviruses from other RNA viruses is the use of a specific host tRNA as a primer to reverse transcribe their genomic RNA (gRNA) into double-stranded proviral DNA. In HIV-1, human tRNA^Lys3^ serves as the primer and is selectively packaged into virions along with the other major tRNA^Lys^ isoacceptors, tRNA^Lys1,2^ [1,2]. Canonically, mammalian tRNAs are virtually always protein-bound [3] and are generally present in the aminoacylated state inside cells [4]. Lysyl–tRNA synthetase (LysRS), the only cellular factor known to interact specifically with all tRNA^Lys^ isoacceptors, is selectively packaged into HIV-1 [5]. LysRS and tRNA^Lys^ are packaged in equimolar proportions, and siRNA knockdown of LysRS resulted in similarly reduced levels (~70%) of tRNA^Lys^ packaging and viral infectivity [6]. Moreover, the packaging of tRNA^Lys^ requires interaction with LysRS [5,7], but not tRNA aminoacylation [8], which would inhibit reverse transcription initiation through occlusion of the required free 3′ hydroxyl. These data support the critical role of LysRS in tRNA^Lys^ packaging; however, the exact mechanism of selecting non-aminoacylated tRNA^Lys3^ remains unclear.

Reports on the primary source of LysRS packaged into HIV-1 have been diverse, leaving this an unsettled question. As a result of alternative splicing, two isoforms of LysRS that differ by only ~20 residues at the N-terminus, cytoplasmic and mitochondrial, are expressed in human cells [9], and both have been reported present in HIV-1 virions [8,10]. In the cytoplasm, two LysRS homodimers are bound to a multi-aminoacyl-tRNA synthetase complex (MSC) through interactions between the dimer interface and the *N*-terminal region of a scaffolding protein known as aminoacyl-tRNA synthetase-interacting protein 2 (AIMP2) [11,12]. Another report suggested that packaged LysRS does not originate from the MSC, nucleus, mitochondria, or the cell membrane; instead, newly-synthesized cytoplasmic LysRS was proposed to interact with HIV-1 Gag before entering any of these compartments [13]. MSC release of LysRS is known to be driven by serine 207 phosphorylation (pS207), which causes a structural change in LysRS to a more open conformation that disrupts the AIMP2 binding site [11,12,14]. We recently reported that HIV-1 infection causes cytoplasmic, MSC-bound LysRS to be phosphorylated on S207, resulting in its release from the MSC and partial re-localization to the nucleus [15]. Treatment with a MEK inhibitor that prevents S207 phosphorylation and nuclear localization reduced viral infectivity by three- to seven-fold [15]. Furthermore, a phosphomimetic variant of LysRS (S207D) also localized to the nucleus and was packaged into HIV-1 virions [15]. In vitro experiments showed that despite retaining wild-type (WT) levels of tRNA^Lys3^ binding, S207D LysRS lacked aminoacylation activity [11,15]. Therefore, selection of the pS207 isoform of LysRS for tRNA^Lys^ packaging may be an adaptation by HIV-1 to ensure that the primer has a free 3′ hydroxyl to initiate proviral DNA synthesis.

The large size (~9.1 kb) and structural complexity [16] of the HIV-1 gRNA has raised questions regarding the mechanism by which tRNA^Lys3^ is released from LysRS and specifically targeted to the primer-binding site (PBS) in the 5′UTR. Just upstream of the PBS lies a tRNA-like element (TLE), which mimics the U-rich anticodon loop sequence of tRNA^Lys3^; this TLE binds preferentially to LysRS [17], promoting the release of tRNA^Lys3^ directly adjacent to the PBS. Nuclear magnetic resonance (NMR) spectroscopy experiments highlighted the strikingly similar interaction of the anticodon binding domain of LysRS with either the TLE or the anticodon stem-loop of tRNA^Lys3^ [18]. SAXS analysis revealed that the global structure of the 105-nt PBS/TLE (PBS/TLE_105_) domain mimics that of the canonical L-shape of tRNA^Lys3^ [19]. The affinity of LysRS for the gRNA is increased in constructs that contain the packaging signal stem-loops SL1 and SL2, compared to only the 105-nt PBS/TLE domain [17], suggesting additional interaction sites outside of the TLE.

Early mechanistic models of LysRS packaging were based on unmodified cytoplasmic LysRS [5,13,17,19,20]. More recently, LysRS phosphorylation has been shown to be important for tRNA-synthetase release from the MSC and tRNA primer packaging [15,21]. Since S207D-LysRS retains the ability to be packaged into HIV-1 virions [15], the phosphomimetic mutation is a good proxy for phosphorylation. In this work, we addressed the open question of how S207 modification affects interaction with the gRNA. We carried out a comprehensive in vitro comparison of WT and S207D-LysRS interactions with the HIV-1 gRNA using an integrated biophysical approach. These results are consistent with a model wherein S207D-LysRS preferentially binds to the PBS of dimeric gRNA in an open conformation that is distinct from that of a non-phosphorylated protein. Additionally, we show that S207D is uniquely capable of destabilizing the PBS region and SL1, a feature that may facilitate correct tRNA primer placement.

## 2. Materials and Methods

### 2.1. Preparation of Recombinant LysRS Variants

Plasmid pCDNA3ΔN65.LysRS [22] was mutated using site-directed ligase-independent mutagenesis [23] to make pCDNA3.207DLysRS [15]. These plasmids, which encode His-tagged human WT LysRS(ΔN65) and S207D LysRS(ΔN65) were transformed into *E. coli* strain BL21(DE3) CP-RIL, and proteins were prepared as follows. Starter cultures were grown overnight in 50 mL of LB media containing 100 µg/mL ampicillin at 37 °C. A 10 mL aliquot of this culture was added to 1 L of LB media containing 100 µg/mL ampicillin and grown for 90 min at 37 °C. After cooling cells to room temperature for 15 min, protein expression was induced using 0.1 mM isopropyl-β-D-1-thiogalactopyranoside (IPTG) and incubated in a room temperature shaker overnight (~16 h). Cells were harvested by centrifugation at 8000× *g* for 15 min at 4 °C followed by resuspension of the cell pellet in 20 mL of lysis buffer (50 mM Na_2_HPO_4_, pH 7.5, 300 mM NaCl, 20 mM imidazole, and 1 protease inhibitor tablet (Roche) per liter). The cells were lysed by sonication for 12 cycles of 15 s of sonication followed by 45 s of rest. Polyethylenimine (0.5% *w/v* final concentration) was added, and the solution was stirred gently for 30 min at 4 °C and cleared of cellular debris by centrifugation at 27,000× *g* for 15 min at 4 °C. Unwanted protein was precipitated by the addition of 375 mg/mL ammonium sulfate, incubated at room temperature for 15 min, and pelleted by centrifugation at 27,000× *g* for 15 min at 4 °C. The pellet was washed with 50 mL of lysis buffer supplemented with 375 mg/mL ammonium sulfate and re-centrifuged at 27,000× *g* for 15 min at 4 °C. The pellet was resuspended in 20 mL of lysis buffer and further centrifuged at 27,000× *g* for 15 min at 4 °C to remove insoluble debris. The supernatant was loaded onto a Ni-NTA (Sigma) column and washed with lysis buffer containing 20 mM imidazole. LysRS constructs were eluted using increasing amounts of imidazole and fractions containing hLysRSΔN65 were combined and concentrated using 30 K molecular weight cut-off centrifugal filter units (Amicon) by centrifuging at 3200× *g* at 4 °C. The concentrated protein was then dialyzed overnight in a 10 K molecular weight cut-off Slide-A-Lyzer dialysis cassette (Thermo Scientific) into 2× storage buffer (80 mM HEPES, pH 7.5, 300 mM NaCl, 4 mM dithiothreitol, and 1 protease inhibitor tablet (Roche) per liter). After dialysis, the protein solution was further concentrated using 30 K molecular weight cut-off centrifugal filter units (Amicon) by centrifuging at 3200× *g* at 4 °C. The concentrated protein solution was diluted with 1 volume of 80% glycerol (40% final) and the concentration was determined using the Bradford assay (Bio-Rad) and stored at −20 °C. The concentration determined by the Bradford assay was multiplied by a correction factor of 1.35 (determined previously by total amino acid analysis) to yield the true concentration. Since hLysRS∆N65 tends to aggregate after extended periods of storage (>6 months), freshly prepared protein was used for all experiments.

### 2.2. Preparation and Purification of RNAs and DNA-Primer-Annealed RNA Complexes

All DNA oligonucleotides used in this study (antiPBS_18_, antiPBS_18_+3, antiPBS_18_+6, and antiPBS_18_+11) were purchased from IDT (Coralville, IA, USA) and used without further purification. All RNAs—tRNA^Lys3^, PBS/TLE, UTR_240_, UTR_240_(ΔDIS), and UTR_240_(ΔDIS, ΔPBS)—were prepared by in vitro transcription using T7 RNA polymerase and purified using denaturing (8 M urea) polyacrylamide gel electrophoresis (PAGE). Human tRNA^Lys3^ was prepared from a plasmid purchased from IDT (pIDTSmart-htRNALys3) containing the tRNA^Lys3^ sequence flanked by a T7 promoter on the 5′ side and a FokI restriction site on the 3′ side. PBS/TLE, corresponding to nucleotides (nt) 126 to 224 of the HIV-1 gRNA with an additional three 5′ G and three 3′ C nucleotides to enable efficient transcription, was transcribed from a FokI-digested plasmid as described [17,19]. UTR_240_ and UTR_240_(ΔDIS), corresponding to nt 106–345 of HIV-1 gRNA, were PCR amplified from plasmids containing the WT sequence or a dimerization-defective mutant, where the dimerization initiation site (DIS) palindrome (5′ AAGCGCGCA 3′) was replaced by a GAGA tetraloop (ΔDIS). The plasmids contained two additional mutations corresponding to U106G and U108C, and the reverse primer contained two mutations corresponding to U342C and G345C. The mutations allowed for efficient transcription by encoding two 5′ guanosines and helped to stabilize the UTR_240_ stem. The PCR products were used as templates for in vitro transcription. To determine an optimal RNA:DNA primer ratio for preparing DNA primer-annealed PBS/TLE, the RNA/DNA complex was heated to 80 °C for 2 min, 60 °C for 4 min, followed by the addition of MgCl_2_ to 1 mM, incubating at 37 °C for 6 min, and cooling on ice for at least 30 min. Complexes were analyzed by native PAGE (Appendix A) and the optimal ratio was determined to be 1:1.5 for PBS/TLE:DNA primer.

### 2.3. Fluorescence Anisotropy (FA) Binding Assays

RNAs used for FA assays were 3′-end labeled with fluorescein-5-semithiocarbazide (FTSC) (Invitrogen) as described [24,25]. Briefly, the RNA (2.5 nmoles) was oxidized using 75 nmoles of NaIO_4_ and then incubated with 2.5 mM FTSC overnight at 4 °C. The free FTSC dye was removed using G-25 Sephadex columns (Roche). The labeling efficiencies were calculated as described [25] and were typically between 75 and 90%.

FTSC-labeled UTR_240_ RNAs were folded in 50 mM HEPES, pH 7.5 by heating at 80 °C for 2 min, 60 °C for 2 min followed by the addition of 10 mM MgCl_2_, and 37 °C for 15 min, then incubating them on ice for at least 30 min. PBS/TLE and DNA-primer-annealed PBS/TLE complexes were prepared with FTSC-labeled PBS/TLE alone or in the presence of a 1.5 molar excess of DNA primer in 50 mM HEPES, pH 7.5 and refolding them using the heating strategy described above with the 37 °C incubation step shorted to 6 min. FA binding assays were performed as previously described [17] with minor modifications. Refolded fluorescently labeled RNAs (20 nM) were mixed with varying concentrations of WT or S207D LysRS(ΔN65) by serial dilution from 10 µM to 16 nM in 20 mM Tris–HCl, pH 8, 15 mM NaCl, 35 mM KCl and 1 mM MgCl_2_. Anisotropy was measured by excitation at 485 nm with the emission read at 525 nm. The data points were fit assuming 1:1 binding [26]. All FA binding data are reported as the average and standard deviation of three independent experiments.

### 2.4. UV Crosslinking with SHAPE (XL-SHAPE) Experiments

Purified in vitro transcribed WT UTR_240_ RNAs in complex with WT and S207D LysRS(ΔN65) were probed by selective 2′-hydroxyl acylation analyzed by primer extension (SHAPE) using *N*-methylisotoic anhydride (NMIA) (Sigma-Aldrich) as described, with minor variations [16,27]. Briefly, RNAs (5 µM) were folded in buffer containing 50 mM HEPES, pH 7.5 and 1 mM MgCl_2_, as described above, and then diluted to 0.5 µM in 20 mM HEPES, pH 7.5, 60 mM NaCl, 10 mM KCl and 1 mM MgCl_2_, as well as different concentrations of the corresponding LysRS variant (2.5, 7.5, 15 and 25 µM). Each reaction (10 µL) was incubated at room temperature for 30 min to allow the binding to reach equilibrium. Reactions were initiated with 1 μL NMIA (80 mM stock in DMSO) or neat DMSO for the (+) and (−) reactions, respectively, and incubated at 37 °C for 30 min followed by phenol–chloroform extraction and ethanol precipitation. For RNA-only controls, protein storage buffer was added instead of protein. The following six reactions were included in each experiment: a control reaction of RNA with DMSO, RNA with NMIA, and RNA with NMIA in the presence of the four noted concentrations of LysRS.

For crosslinking analyses, RNA:LysRS complexes were prepared as described for SHAPE. Samples (10 µL) were crosslinked on ice in a Stratalinker 2400 UV crosslinker (Stratagene) using a total energy of 200 mJ/cm^2^. Non-crosslinked control samples were incubated on ice for the same amount of time. The following six reactions were included in each experiment: a no protein/no UV background, RNA only with crosslinking, and RNA incubated with the four different concentrations of LysRS noted above. Following the crosslinking reactions, protein was denatured with 1 µL of 5% SDS (0.5% final) followed by the addition of 1 µL of Proteinase K (800 U/mL, New England Biolabs, Ipswich, MA, USA) and incubation at 55 °C for 60 min. RNA was recovered by phenol–chloroform extraction and ethanol precipitation.

RNA samples from both SHAPE and crosslinking experiments were resuspended in 9 µL RNase-free water followed by the addition of dNTP mix (1 µL of 10 mM each) and 2 µL of 5 µM 5′-NED^TM^ labeled primer (Applied Biosystems, Waltham, MA, USA; Life Technologies, Carlsbad, CA, USA) and incubation at 85 °C for 1 min, 60 °C for 5 min, 35 °C for 5 min and 50 °C for 10 min. Reverse transcription mix (8 µL) containing 1 µL of SuperScript III (200 U/µL), 4 µL 5× first-strand buffer, 2 µL 0.1 M DTT and 1 µL MiliQ H_2_O was then added. Samples were incubated at 55 °C for 1 h and inactivated at 70 °C for 15 min following the manufacturer’s protocol (Invitrogen). RNA was hydrolyzed with 1 µL of 4 M NaOH and incubated at 95 °C for 3 min. Samples were cooled on ice and neutralized with 2 µL of 2 M HCl. A description of the primers used for these experiments can be found in Appendix A.

Dideoxy sequencing reactions were conducted on the same DNA template used for the in vitro transcription of the UTR_240_ constructs, using the Thermo Sequenase Cycle Sequencing Kit (Affymetrix) and the same NED^TM^-labeled primers as those used for primer extension (Appendix A). Sequencing reactions containing 200 ng DNA template and 1 pmol NED^TM^-labeled primer were heated to 95 °C for 1 min followed by 60 cycles of 95 °C for 30 s, 55 °C for 30 s and 72 °C for 1 min in a T100 thermal cycler (Bio-Rad). Sequencing reactions were performed independently with each NED^TM^-labeled primer. Primer extension products from the SHAPE experiments, crosslinking experiments and sequencing reactions were analyzed by capillary electrophoresis on an Applied Biosystems 3730 DNA Analyzer (Genomics Shared Resources Facility, The Ohio State University, Columbus, OH, USA) after resuspension of the cDNA pellets in formamide and mixing with GeneScan™ 600 LIZ^®^ Size Standard (Applied Biosystems, Waltham, MA, USA) for inter-capillary alignment as described [28].

Raw electropherograms were converted to normalized SHAPE reactivity using the ribonucleic acid capillary–electrophoresis analysis tool (RiboCAT) [28]. Peak areas for each of the six reactions were calculated and then scaled to the RNA-only background reaction. At least three independent experiments were performed and the data from each independent trial were analyzed individually to generate average SHAPE reactivities. The differences between the SHAPE reactivity of the free RNA and that of the reaction in the presence of the highest protein concentration were analyzed by an unpaired two-tail Student’s *t*-test. Statistical significance was attributed to peaks characterized by an absolute difference in SHAPE reactivity of ≥ 0.3 arbitrary units and a *p*-value < 0.05 between the RNA-only control and the highest protein concentration sample [29]. In addition, only those peaks for which a dose-dependence could be observed were deemed significant.

### 2.5. Small-Angle X-ray Scattering (SAXS) Analysis

The UTR_240_(ΔDIS) and UTR_240_(ΔDIS, ΔPBS) constructs (400 µg) were refolded in a volume of 200 µL by heating in 50 mM HEPES, pH 7.4, to 80 °C for 2 min, 60 °C for 2 min, followed by addition of 10 mM MgCl_2_. The samples were then incubated at 37 °C for 15 min, and on ice for at least 30 min. The folded RNAs were purified by size exclusion chromatography (SEC) on a 24-mL Superdex 200 10/300 GL Increase column (GE Healthcare, Chicago, IL, USA) using a GE ÄKTA purifier in a buffer containing 150 mM NaCl, 50 mM HEPES, pH 7.4, 1 mM MgCl_2_, and 3% glycerol (*wt/vol*) at a flow rate of 0.5 mL/min. The absorbance of the eluate was monitored at 260 nm. Fractions corresponding to the peak of interest were concentrated to ~40 µL and serially diluted to yield three different concentrations for the analysis of each sample. Prior to shipment to the SIBYLS beamline at Lawrence Berkeley National Labs for data collection [30], an aliquot of the highest concentration sample was retained for native PAGE analysis to be performed at the same time the SAXS data were to be collected [31]. There were no signs of degradation or heterogeneity by native PAGE.

SAXS data analysis steps were carried out as previously described [19,31] using the program PRIMUS [32]. Guinier analysis was used to calculate the R_g_ of the RNA, and Kratky analysis was performed to determine the extent of folding [31]. If the Guinier plot displayed non-linearity (a sign of poor-quality data) or Kratky analysis indicated that the RNA was not well-folded, then the data were not analyzed further. The P(r) function was calculated using the program GNOM [32], and the D_max_ and P(r)-based R_g_ was determined using the AUTOGNOM feature. Ab initio envelopes were generated largely as previously described using the ATSAS software suite [19,31,32]. Briefly, 50 ab initio envelopes were generated in jagged mode with no symmetry restraints imposed, and the χ^2^ fits and reproducibility (NSD) values were determined. These 50 envelopes were averaged into one envelope, which was then packed with additional “dummy atoms” with an atomic radius of 2.0 Å. This was then used as the starting point for an additional 50 ab initio envelope calculations, generated in expert mode with no symmetry restraints imposed. These envelopes were averaged and filtered to generate the final envelope, and their χ^2^ fits and NSD values were calculated.

### 2.6. SEC–SAXS Analysis of Primer-Annealed and LysRS Bound PBS/TLE Complexes

Size-exclusion chromatography coupled with SAXS (SEC–SAXS) was used to analyze complexes to alleviate problems with sample heterogeneity. Each of the DNA primer-annealed PBS/TLE RNA complexes (PBS/TLE annealed to each of the antiPBS_18_, antiPBS_18_+3, antiPBS_18_+6, and antiPBS_18_+11 DNA primers) were prepared using 1 mg total nucleic acid with a 1.5 molar excess of DNA primer and refolded in a volume of 200 µL by heating the sample in 50 mM HEPES, pH 7.4, to 80 °C for 2 min, 60 °C for 4 min, followed by the addition of 10 mM MgCl_2_. The sample was then incubated at 37 °C for 6 min and on ice for at least 30 min. For LysRS-bound PBS/TLE complexes, PBS/TLE was first refolded as described for the primer-annealed PBS/TLE complexes and purified by SEC on a 24-mL Superdex 200 10/300 GL Increase column (GE Healthcare, Chicago, IL, USA) using a GE ÄKTA purifier in a buffer containing 150 mM NaCl, 50 mM HEPES, pH 7.4, 1 mM MgCl_2_, and 3% glycerol (*wt/vol*) at a flow rate of 0.5 mL/min. The absorbance of the eluate was monitored at 260 nm. Fractions corresponding to the peak of interest were concentrated to ~40 µL. Purified PBS/TLE was then incubated with a 1:1 molar ratio of either WT or S207D LysRS for 30 min prior to SEC-SAXS analysis.

SEC–SAXS data were collected at station G1 at the Cornell High Energy Synchrotron Source (CHESS) using the experimental set-up described [33]. Briefly, refolded samples were loaded into a GE ÄKTA Pure M with a 24-mL Superdex 200 10/300 GL Increase column (GE Healthcare, Chicago, IL, USA) and SEC was run in 150 mM NaCl, 50 mM HEPES, pH 7.4, 1 mM MgCl_2_, and 3% glycerol (*wt/vol*) at a flow rate of 0.5 mL/min. The absorbance of the eluate was monitored at 260 nm. The ÄKTA Pure M is coupled to the beamline such that the sample goes directly from the absorbance detector into a flow-cell in the beamline. The X-ray source was a 1.5 m CHESS Compact Undulator, and X-rays were detected using a Finger Lakes charged–coupled device (CCD). The beam diameter was 250 × 250 µm with a flux of 8.4 × 10^11^ photons s^−1^ and an energy of 9.962 keV. X-ray exposures were taken continuously every 4 s while the SEC eluent passed through the SAXS flow cell. Initial SAXS data processing, including radial averaging and buffer subtraction, was performed using the *BioXTAS RAW* software program [34]. SAXS exposures (10 × 4 s) corresponding to the RNA peak of interest on the SEC were averaged, and 50 × 4 s SAXS exposures corresponding to the buffer alone were averaged and then used to buffer subtract the RNA average SAXS curve [35]. Average SAXS curves were analyzed as described above. The envelope for PBS/TLE alone was recalculated as described above for SAXS analysis using data from a previously published study [19].

## 3. Results

### 3.1. The DIS Contributes to gRNA Binding of Both WT and S207D LysRS(ΔN65)

A LysRS(ΔN65) variant with the *N*-terminal 65 residues deleted is more sensitive to specific RNA binding interactions [36], and a truncated form of LysRS in virions likely contains this deletion [5]. Therefore, all experiments described for both the WT and the phosphomimetic S207D mutant were performed in the context of LysRS(ΔN65) (Figure 1A). Previous work investigating LysRS–gRNA binding were performed using WT LysRS(ΔN65). This protein specifically binds to the TLE motif [17,18,19], but higher-affinity binding of LysRS(ΔN65) to gRNA additionally required regions of the Psi RNA-packaging domain [17,37]. Therefore, to determine whether S207D LysRS(ΔN65) retains the same binding characteristics, FA binding assays were performed using 3′ fluorescently labeled RNAs corresponding to in vitro transcribed human tRNA^Lys3^, the 105-nt PBS/TLE_105_ domain alone or constructs (UTR_240_ and UTR_240_(ΔDIS)) containing both the PBS/TLE and Psi domains. The first nine nucleotides of the gag coding region were included because they contribute to the structure and stability of the packaging signal [38], but the first 104-nt (TAR and polyA hairpins) were excluded because they do not appreciably contribute to LysRS binding [17]. Native PAGE analysis of the WT UTR_240_ RNA showed complete dimerization under the conditions used for FA assays (Appendix A). The ΔDIS variant contains a mutated DIS that eliminates gRNA dimerization (Figure 1B). To promote transcription efficiency and stabilize the terminal stems, the first U–G base pair was changed to a G–C pair and the two following U–G/G–U wobble pairs were mutated to C–G/G–C pairs, respectively (Figure 1A). Both WT and S207D LysRS(ΔN65) bound tRNA^Lys3^ and PBS/TLE_105_ with similar affinity but bound the Psi-containing WT UTR_240_ with approximately two-fold higher affinity (Table 1, Appendix A). This closely matches our previous findings for WT LysRS(ΔN65) [17]. Mutation of the DIS palindrome to a GNRA tetraloop (Figure 1B) reduced affinity back to the level of tRNA^Lys3^ and PBS/TLE_105_ for both LysRS(ΔN65) variants. Therefore, the higher binding affinity is either due to specific interactions with the DIS sequence or gRNA dimerization/conformational differences.

### 3.2. S207D LysRS(ΔN65) Binds the PBS/TLE in an Open Conformation

Results of the binding assays suggest that WT and S207D LysRS(ΔN65) have similar RNA binding affinities, displaying somewhat higher affinity binding to the dimeric gRNA. SAXS was next used to investigate the structures formed upon WT and S207D LysRS(ΔN65) binding to the PBS/TLE RNA. SAXS yields ab initio molecular envelopes that reflect the global fold of the molecule or complex with a calculated maximum dimension (D_MAX_). Signs of aggregation, radiation damage or heterogeneity were absent from the obtained SAXS data (Appendix A). When compared to PBS/TLE alone (D_MAX_ = 123 Å), the binding of WT and S207D LysRS(ΔN65) resulted in more extended envelopes with D_MAX_ of 139 and 162 Å, respectively (Figure 2). The additional envelope density for both complexes corresponds to the region previously mapped to the TLE stem and loop [19]. A previous SAXS analysis of LysRS(ΔN65) reported that both WT and S207D variants form stable homodimers, with S207D adopting a more open conformation resulting from a 37-degree rotation of the N-terminal domains of each monomer with respect to their respective catalytic domains [11].

### 3.3. S207D LysRS(ΔN65) Binding Induces gRNA Conformational Flexibility

To characterize both WT and S207D LysRS(ΔN65) binding sites and RNA conformational changes upon protein binding, XL-SHAPE studies were carried out using the WT UTR_240_ with the capability to dimerize. SHAPE is a single-nt resolution RNA structure probing technique where anhydride SHAPE reagents react with the 2′-hydroxyl of all four nts to form covalent adducts that stop primer extension by reverse transcriptase [39,40]. SHAPE reagents are more reactive to less constrained, unpaired nts [39,41]. SHAPE probing of protein–RNA complexes can be used to identify RNA conformational changes associated with protein binding but cannot reliably identify direct binding sites [42]. To identify binding sites, UV-crosslinking of the protein:RNA complex was carried out. Protease digestion following UV irradiation leaves a covalent adduct on the RNA that can similarly be characterized by primer extension stops. For both SHAPE and XL experiments, a protein titration was performed with stringent criteria for positives indicated by a dose-dependent reactivity change of >0.3 from the “no protein” control to the highest protein concentration (Appendix A).

The baseline SHAPE reactivity pattern of the UTR_240_ (Figure 3) closely matches results that we and others have previously reported [16,27,28]. In agreement with binding data (Table 1), clusters of XL sites were identified in the TLE and SL1 hairpins for both WT and S207D LysRS(ΔN65), with very similar XL footprints (Figure 3, asterisks). Three XL sites were also identified in the unstructured PBS loop upon titration of S207D LysRS(ΔN65), but not WT. The more flexible “open conformation” of the S207D LysRS(ΔN65) [11] may have facilitated the interaction with the PBS loop when compared to the more compact WT protein. Consistent with the binding assays showing higher affinity binding to an RNA construct that also contains SL1, both LysRS(ΔN65) constructs directly bind the SL1 stem of the UTR240. Although a higher affinity binding requires the DIS sequence (Table 1), no crosslinks were identified in the palindromic DIS. Only the S207D variant showed XL sites in and proximal to the DIS loop, which could be due to this protein binding in the more open conformation.

Despite their similar binding affinity and XL footprints, a larger number of SHAPE reactivity changes resulted from S207D LysRS(ΔN65) binding when compared to WT LysRS(ΔN65) (25 and 3, respectively). Indeed, 22 of the 25 differences upon titration of S207D LysRS(ΔN65) are increases in SHAPE reactivity, located predominantly in the PBS loop and SL1 stem (Figure 3, red arrows), indicating an increase in the conformational flexibility of these regions. The binding of both proteins resulted in reduced flexibility (Figure 3, blue arrows) at the 3′ side of the TLE loop, likely because of protein binding.

### 3.4. The Monomeric UTR_240_(ΔDIS) Adopts an Extended Cruciform-like Conformation

The binding assays and XL data suggest that both LysRS(ΔN65) constructs directly bind the TLE and SL1 stems of the UTR_240_ and do not interact directly with the DIS sequence. SAXS was next used to determine the overall tertiary fold of the UTR_240_(ΔDIS) gRNA construct. The SAXS data showed no signs of aggregation, radiation damage or heterogeneity (Appendix A). The resulting SAXS-derived ab initio molecular envelope revealed a cruciform-like shape (Figure 4A). Furthermore, one appendage has a distinct L-shaped structure expected of the PBS/TLE domain [19]. To map the region of the envelope corresponding to the PBS/TLE, we also performed SAXS analysis of a truncated 145-nt UTR_240_(ΔDIS, ΔPBS/TLE) RNA with a GAGA tetraloop inserted in place of the PBS/TLE domain. PBS/TLE truncation resulted in a structural model wherein the L-shaped region is missing from the ab initio envelope (Figure 4B, blue spheres). A structure corresponding to the UTR_240_(ΔDIS) with the PBS/TLE domain replaced with a tetraloop, termed the “core encapsidation signal”, was previously determined by NMR [38]. To assign the remaining domains, we overlayed the SL1, SL3 and U5:AUG stem loops from the reported NMR structure into our SAXS-derived model using the size and shape of each stem loop as a guide (Figure 4C). We found excellent agreement with one cruciform arm and SL1 domain from the NMR structure (Figure 4C, orange), and were also able to fit the remaining helices from the NMR structure into our envelope (Figure 4C, red, yellow, purple).

Based on our SAXS data, we conclude that the TLE and SL1 are oriented in opposite directions in the monomeric UTR_240_(ΔDIS). This orientation is too distant for a single WT or S207D LysRS(ΔN65) homodimer (Figure 4D) to bind both sites without a significant conformational change. Importantly, the SAXS envelope indicates a structure in which SL1 is properly folded, with the DIS exposed in the expected dimer-competent conformation. Therefore, the reduced binding of LysRS(ΔN65) to the UTR_240_(ΔDIS) relative to the WT UTR_240_ is not due to misfolding of the RNA. Rather, tighter binding of both LysRS(ΔN65) variants to the UTR_240_ requires gRNA dimerization.

### 3.5. Disruption of TLE Structure by Primer Extension Releases Bound LysRS

Previous work demonstrated that the TLE hairpin is the primary determinant for LysRS(ΔN65) binding to gRNA via anticodon loop sequence mimicry and structure mimicry of the canonical tRNA L-shaped fold [18,19,37]. In our proposed model, LysRS facilitates correct primer localization by releasing tRNA^Lys3^ in close proximity to the PBS due to competition by the TLE for LysRS binding [17,19]. Therefore, because LysRS binding to the TLE precedes primer annealing, we hypothesized that the initiation of reverse transcription will induce conformational changes in the TLE that facilitates the release of LysRS, ensuring that LysRS binding does not significantly inhibit proviral DNA synthesis.

To test this hypothesis and characterize the conformational changes that occur during reverse transcription initiation, binding assays were performed between WT and S207D LysRS(ΔN65) variants and PBS/TLE constructs annealed to short DNA oligonucleotides resembling different stages of initial reverse transcription products (Figure 5A). The annealing of antiPBS_18_ and antiPBS_18_+3 should not significantly alter the TLE structure, but annealing of antiPBS_18_+6 and antiPBS_18_+11, with complementary regions that extend into the TLE stem, would be expected to cause gradual unfolding of the TLE hairpin. Reverse transcription pauses were observed at +3 and +5 sites upon primer extension, due to the stability of the TLE stem [43]. As expected, binding of LysRS(ΔN65) was relatively unaffected by a 3-nt extended primer, but further extension progressively increased the dissociation constant for both WT and S207D LysRS(ΔN65), consistent with destabilization of the complex (Table 2, Appendix A). The +6 extended primer had reduced affinity despite not annealing to any nt that directly crosslink to either LysRS(ΔN65) variant (Figure 3). Similarly, the +11 primer only annealed to 2 or 1 nt that crosslinked to WT or S207D LysRS(ΔN65), respectively.

We previously proposed [17,18,19] that the overall conformation of the TLE region is likely to play a role in specific LysRS(ΔN65) recognition. SAXS was next used to characterize the structural consequences of extended-primer annealing to the PBS/TLE domain. For these experiments, SEC–SAXS was used, wherein primer-annealed PBS/TLE complexes were separated by SEC immediately prior to SAXS analysis [31,33,34,44]. The primer was present in a 1.5× molar excess to ensure that close to 100% of the PBS/TLE was in the annealed state. All data collected for the PBS/TLE and primer annealed complexes lacked any signs of heterogeneity, aggregation or interparticle interactions/repulsions (Appendix A) [31]. The conformation of the PBS/TLE RNA with antiPBS_18_ annealed (Figure 5B) closely resembled our previously reported structure [19]. SAXS envelopes for antiPBS_18_+3, antiPBS_18_+6 and antiPBS_18_+11 annealed complexes show that the TLE has a progressive counter-clockwise rotation of ~65°, ~145° and ~225°, respectively, with respect to the planar PBS region (Figure 5C, Appendix A and Table 2).

## 4. Discussion

Retroviruses have limited sequence space to encode all biological requirements for efficient replication. To maximize functionality, host factors are co-opted to facilitate replication. HIV-1 must package a non-aminoacylated tRNA^Lys3^ primer to initiate reverse transcription and has evolved to selectively package a complex of phosphorylated LysRS, which lacks aminoacylation capability, and its cognate tRNA [15]. Previously, we proposed that competition from the gRNA 5′UTR for LysRS binding facilitates tRNA^Lys3^ release from LysRS, which interacts with a TLE stem loop directly adjacent to the PBS [17,18,19,37]. S207D LysRS(ΔN65) is known to exist primarily as a homodimer with a conformation that is more “open” when compared to WT LysRS [11]. It was unclear whether this structural switch has any effect on the 5′UTR binding properties or the mechanism of LysRS-facilitated primer placement. The similarities in binding constants and crosslinking sites reported here suggest that S207D LysRS(ΔN65) retains the characteristic binding properties of WT LysRS(ΔN65) (Table 1, Figure 3). Indeed, both LysRS variants have a two- to three-fold preference for binding to 5′UTR constructs that contain the Psi packaging signal with an intact DIS (Table 1). Importantly, based on crosslinking data, neither LysRS variant directly binds to the DIS. We therefore conclude that the competition for LysRS binding that results in tRNA release proximal to the PBS is more efficient on dimeric HIV-1 gRNAs, the isoform that is packaged into progeny virions [45]. This may help to ensure that phosphorylated LysRS is not diverted from the packaged gRNA dimers to the non-packaged monomeric HIV-1 viral RNAs that serve as mRNAs for viral protein synthesis.

SAXS analysis of LysRS bound to PBS/TLE_105_ revealed ~16 and ~39 Å longer envelopes for WT and S207D LysRS-bound complexes, respectively, relative to the RNA alone (Figure 2). Due to the low-resolution nature of SAXS and the lack of well-defined features in the extra density, it cannot be conclusively determined whether LysRS is binding as a monomer or its more preferred dimer. Attempts to ascertain the stoichiometry of these complexes using native mass spectrometry were unsuccessful due to the aggregation of the complexes in the buffer conditions required for analysis. Nevertheless, a previous study using hydrogen–deuterium exchange mass spectrometry and SAXS reported that both WT and S207D LysRS exist primarily as homodimers in solution [11]. Therefore, it is most likely that both WT and S207D bind to the HIV-1 gRNA as homodimers. Moreover, the larger excess density observed in the S207D LysRS: PBS/TLE_105_ complex is likely caused by the “open conformation” that was previously reported for S207D LysRS homodimers [11].

The other major difference observed between WT and S207D LysRS binding to HIV-1 gRNA was an increased flexibility in the PBS loop and SL1 stem upon binding of S207D LysRS, but not WT (Figure 3). While it remains unclear whether this has any effect on HIV-1 replication, we speculate that it may play a role in facilitating primer annealing or reverse transcription initiation. While not necessary in the NL4.3 isolate used in this study, ~14% of HIV-1 isolates including the well-studied MAL isolate require substantial conformational rearrangement of the PBS region for formation of the reverse transcription initiation complex [46]. While the authors of that study suggested that the HIV-1 nucleocapsid protein facilitates this rearrangement, our results suggest that S207D LysRS may also participate in these conformational changes.

The fact that LysRS binds to both the PBS/TLE and SL1 suggests that these regions of the 5′UTR may be in close proximity in the three-dimensional structure. However, SAXS data revealed that the UTR_240_(ΔDIS) RNA construct adopts a cruciform-like structure with the PBS/TLE and SL1 on opposite ends of the structure (Figure 4). Based on the relative size of LysRS (Figure 4A,B are at the same scale), a single LysRS dimer is not large enough to bind both the PBS/TLE and SL1 without a conformational change in the RNA. Further, the individual stems (PBS/TLE, SL1/SL2, SL3 and U5:AUG) appear to be independently folded and correspond well to those domains from a recently published NMR structure of the core encapsidation signal, which contains the same RNA regions with a truncated PBS/TLE [38]. Although the individual stems fit nicely into the SAXS envelope, the relative orientations of the stems do not match the reported NMR structure. This may be due to conformational differences resulting from inclusion of the PBS/TLE domain, different experimental conditions, or lack of long-range NMR constraints and/or residual dipolar coupling data to obtain orientation data for the helices in the NMR study.

The first secondary structure element that reverse transcriptase encounters during minus-strand strong-stop cDNA synthesis is the TLE stem [47], which represents a potential roadblock if LysRS is tightly bound. Therefore, LysRS must unbind from the TLE during tRNA primer extension for efficient reverse transcription. Our FA data indicate that the affinity of LysRS for the TLE is reduced as a DNA primer is extended into the first 8 nt of the TLE stem (Table 2), likely due to disruption of the tRNA-like conformation of the PBS/TLE domain (Figure 5). This result supports our proposed mechanism of LysRS-dependent primer placement based on tRNA mimicry of the PBS/TLE [19]. This also suggests that the affinity of the PBS/TLE for LysRS is finely tuned to allow for proper primer placement, but not tight enough to provide a substantial barrier to reverse transcription.

Based on these results, a new working model is proposed (Figure 6). In this model, HIV-1 infection triggers phosphorylation of LysRS at S207, resulting in LysRS release from the MSC in a conformation that is competent for tRNA binding but not for aminoacylation [11,15] (Figure 6, Step 1). The catalytically inactive dimeric pS207–LysRS/tRNA complex is then recruited to sites of viral assembly via interactions with HIV-1 Gag [5,48]. The 5′UTR can adopt various monomeric and dimeric conformations, including a “kissing loop” dimer [49,50,51,52,53,54] (Figure 6, Step 2). The anticodon-like TLE of an HIV-1 gRNA dimer then competes for binding to pS207–LysRS resulting in an increase in dynamics of the gRNA in this region and release of the tRNA primer proximal to the PBS (Figure 6, Step 3). Crosslinking and dynamic changes are also observed in the SL1 hairpin, but whether a second LysRS dimer binds directly to this region as shown here remains to be confirmed. The tRNA primer is subsequently annealed to the PBS via chaperone activity of the nucleocapsid domain of the HIV-1 Gag polyprotein [55,56] (Figure 6, Step 4). After viral budding and maturation, reverse transcriptase specifically binds the tRNA:gRNA duplex (Figure 6, Step 5) and initiates the synthesis of minus-strand strong stop proviral DNA from the 3′ end of the uncharged tRNA primer (Figure 6, Step 6). During the initial extension of the primer (3–11 nt), the conformation of the TLE changes and its tRNA mimicry is lost, triggering release of pS207–LysRS from the gRNA. Based on this model, we propose that pS207–LysRS initially selects and facilitates tRNA annealing on the kissing loop dimer subpopulation of gRNA conformers. Annealing of the tRNA primer further stabilizes the kissing dimer and mature nucleocapsid protein subsequently promotes the formation of a more stable extended dimer conformation [49].

## Figures and Tables

**Figure 1 viruses-14-01556-f001:**
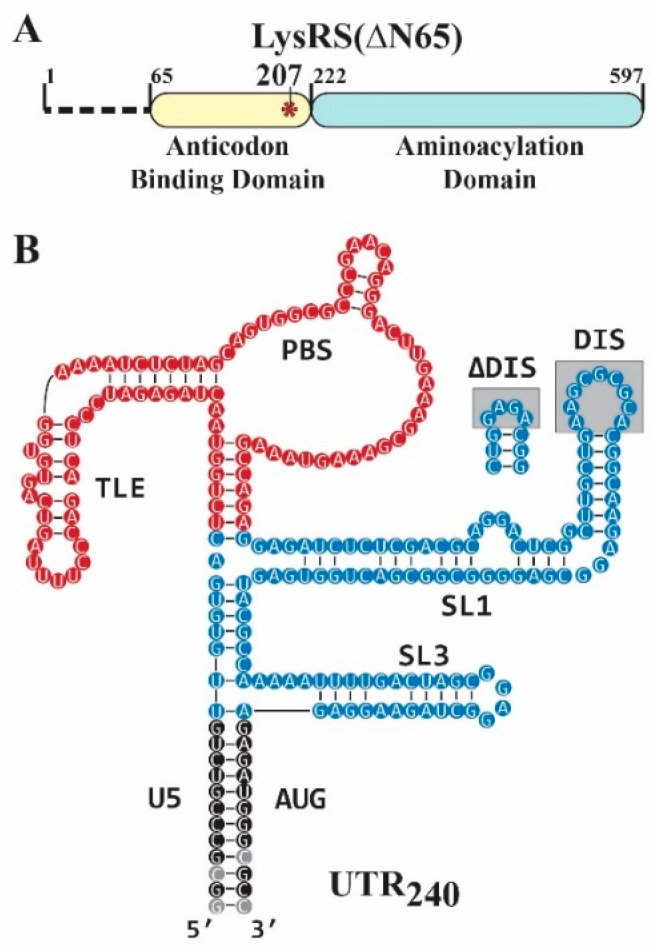
Proteins and RNAs used in this study. (**A**) The 532-residue human LysRS construct used for all experiments lacks the *N*-terminal 65 residues (LysRS(∆N65)). The phosphomimetic mutant replaces a serine at position 207 with an aspartate in the anticodon binding domain (S207D, red asterisk). (**B**) The UTR_240_ construct contains the U5/AUG stem (black), PBS/TLE domain (red, nt 126–224), and Psi domain (blue, nt 117–125 and 225–332). The gray boxed regions indicate the mutation that was made to replace the DIS with a stable GNRA tetraloop (∆DIS). The PBS/TLE domain was also deleted for some studies (ΔPBS/TLE) and replaced with GAGA. Nt in gray circles indicate mutations that were made to promote transcription efficiency and stabilize the terminal stem, as described in the Methods. The secondary structure shown is based on Ref [38].

**Figure 2 viruses-14-01556-f002:**
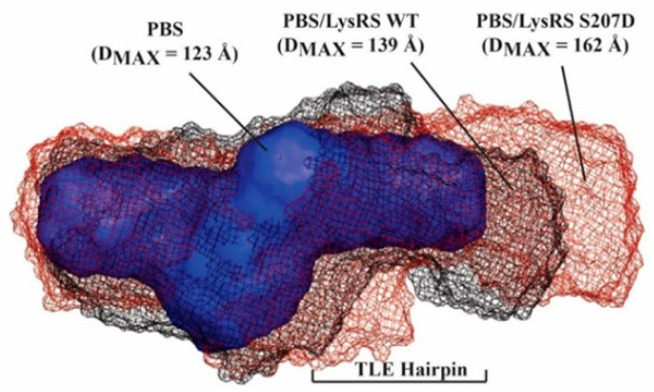
SAXS analysis of LysRS-bound PBS/TLE complexes. The SAXS-derived ab initio envelope for PBS/TLE alone (blue) was overlaid with the envelopes calculated for PBS/TLE bound to either WT LysRS(∆N65) (grey mesh) or S207D LysRS(∆N65) (brown mesh).

**Figure 3 viruses-14-01556-f003:**
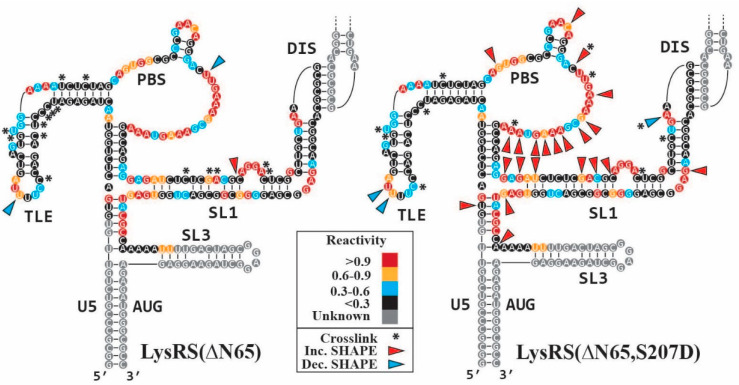
XL-SHAPE results for LysRS(∆N65) variants binding to WT UTR_240_. The baseline SHAPE reactivities are shown as colored circles behind each nt. Grey circles indicate the region of the structure that was not probed in our studies. SHAPE reactivity changes that occurred upon titration of WT LysRS(∆N65) (**left**) and S207D LysRS(∆N65) (**right**) are identified with colored arrows (red = increased reactivity, blue = decreased reactivity) and crosslinked sites are denoted with asterisks.

**Figure 4 viruses-14-01556-f004:**
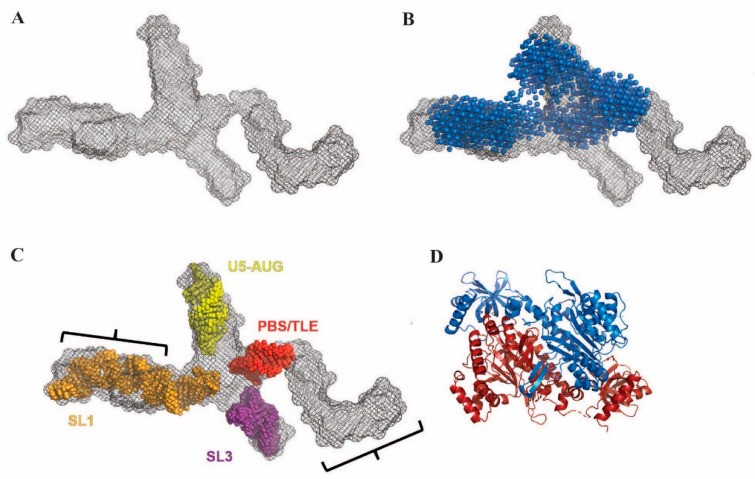
SAXS analysis of the UTR_240_(ΔDIS). (**A**) SAXS envelope of the UTR_240_(ΔDIS) RNA (grey mesh). (**B**) Alignment of the UTR_240_(ΔDIS) with the envelope for UTR_240_(ΔDIS,ΔPBS/TLE) (blue spheres). (**C**) Each of the individual helices (PBS/TLE, red; U5-AUG, yellow; SL1, orange; SL3, purple) from a previously reported NMR structure [38] fit into specific regions of the UTR_240_(ΔDIS) SAXS envelope. (**D**) LysRS dimer crystal structure depicted at the same scale as the SAXS data with one monomer in red and the other in blue. Black brackets indicate approximate locations of LysRS binding sites determined by crosslinking.

**Figure 5 viruses-14-01556-f005:**
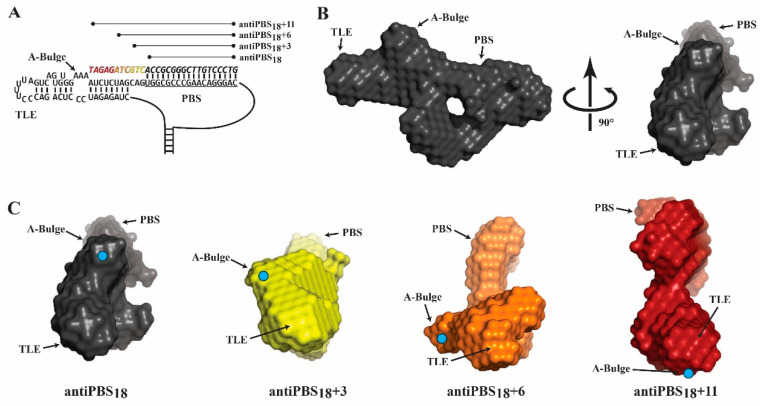
SAXS analysis of extended primer complexes. (**A**) Schematic of construct design for annealing of extended DNA primers (italic font) designed to disrupt the structure of the TLE stem. (**B**) SAXS-derived molecular envelope of the antiPBS_18_-annealed PBS/TLE construct. (**C**) SAXS-derived molecular envelopes for the PBS/TLE RNA with DNA primers of different lengths annealed. When aligned using the PBS loop as a reference, the location of the A-Bulge (cyan dot) rotates counterclockwise about the structure.

**Figure 6 viruses-14-01556-f006:**
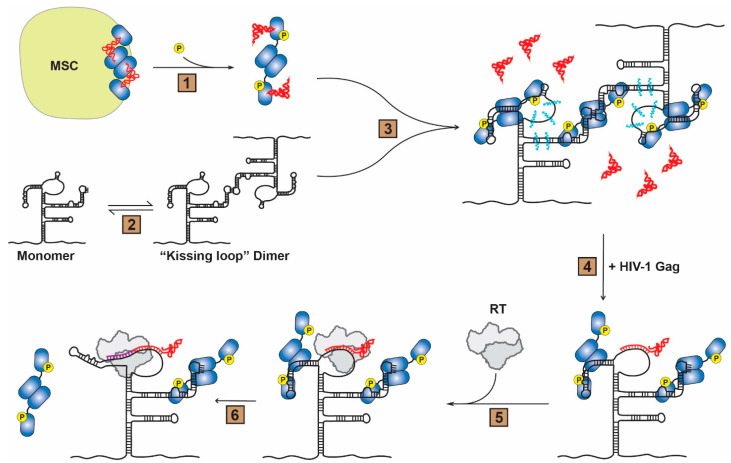
Model of pS207–LysRS-directed primer placement to PBS and subsequent pS207–LysRS release from HIV-1 gRNA. (Step 1) Upon HIV-1 infection, LysRS is phosphorylated on S207 and is released from the multi-aminoacyl-tRNA synthetase complex (MSC) in a conformation inactive for tRNA aminoacylation. (Step 2) The HIV-1 gRNA can adopt various monomeric and dimeric conformations, including the “kissing loop” dimer shown. (Step 3) The catalytically inactive, tRNA-bound pS207–LysRS dimer preferentially binds to the PBS/TLE and SL1 regions of an HIV-1 gRNA dimer. This binding results in release of tRNA from LysRS and an increase in the dynamics of the PBS region and SL1 hairpin (blue squiggles). (Step 4) HIV-1 Gag facilitates annealing of the tRNA primer onto the PBS via chaperone activity of the nucleocapsid domain. (Step 5) After maturation, reverse transcriptase (RT) binds the primer:template complex and (Step 6) initiates proviral DNA synthesis (purple extension) leading to disruption of the TLE conformation and pS207–LysRS release. For clarity, the second gRNA monomer is not shown after Step 4.

**Table 1 viruses-14-01556-t001:** Dissociation constants (K_d_) for WT and S207D LysRS(∆N65) binding to HIV-1 RNAs.

	K_d_ (nM ± SD)	
RNA	WT LysRS(ΔN65)	S207D LysRS(ΔN65)
tRNA^Lys3^	407 ± 33 ^a^	470 ± 190 ^b^
PBS/TLE_105_	330 ± 115	315 ± 130
UTR_240_	146 ± 47	139 ± 45
UTR_240_(ΔDIS)	315 ± 142	482 ± 103

All measurements were performed with 30 nM RNA in 15 mM NaCl, 35 mM KCl, 20 mM Tris–HCl pH 8, and 1 mM MgCl_2_. Results are the average of at least three trials with the standard deviation (SD) indicated. Binding data from Ref [17] ^a^ and Ref [15] ^b^.

**Table 2 viruses-14-01556-t002:** Equilibrium dissociation constants (K_d_) for human WT LysRS(∆N65) and S207D LysRS(∆N65) binding PBS/TLE_105_:antiPBS complexes.

RNA-DNA Complex		K_d_ (nM ± SD)		
RNA	DNA Primer	WT LysRS(ΔN65)	S207D LysRS(ΔN65)	TLE Rotation
PBS/TLE_105_	antiPBS_18_	240 ± 95	417 ± 90	0°
PBS/TLE_105_	antiPBS_18_+3	488 ± 171	305 ± 75	~65°
PBS/TLE_105_	antiPBS_18_+6	865 ± 379	590 ± 182	~145°
PBS/TLE_105_	antiPBS_18_+11	1008 ± 639	949 ± 148	~225°

All binding measurements were performed with 30 nM RNA in 15 mM NaCl, 35 mM KCl, 20 mM Tris–HCl pH 8, and 1 mM MgCl_2_. Results are the average of at least three trials with the SD indicated. All TLE rotation values are approximated by orienting the SAXS envelopes as shown in Figure 5C and drawing a line from the A-bulge to the TLE loop regions of the envelopes. The lines were then compared to reflect the approximate counterclockwise rotation.

## Data Availability

All data collected for this study will be made available upon request.

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
