# Peer review of "Phosphomimetic S207D Lysyl–tRNA Synthetase Binds HIV-1 5′UTR in an Open Conformation and Increases RNA Dynamics"

_viruses, 2022, doi:10.3390/v14071556_

Round 1

Reviewer 1 Report

Cantara et al., presents a comprehensive study of the interactions between the host Lysyl-tRNA synthetase (LysRS) and a HIV-1 5’-UTR region, including the primer-binding site (PBS), PBS-proximal TLE (tRNA-like element), and packaging signal stem loop 1 (SL1). Combining crosslinking, chemical probing, phosphomimetics, SAXS, fluorescence anisotropy, and structure modeling, the authors discovered that LysRS phosphorylation led to a more open conformation of the UTR when bound to LysRS, and enhanced flexibilities of the PBS and SL1 regions, which may facilitate the annealing of the tRNA primer. Based on crosslinking results, the authors propose a revised structural model, where an extended phospho-LysRS dimer binds bivalently to a HIV-1 UTR dimer via the distant TLE and SL1 interfaces. Since phosphorylated LysRS is the functional form that is packaged in virions, this study provides important new insights into the mechanisms of retroviral replication. The manuscript is expertly written, clearly illustrated, technically rigorous, and is both thorough and thoughtful. I enthusiastically support the publication of this outstanding study, and have but a few minor, largely cosmetic suggestions below.

Specific comments:

1.      Since the authors already include raw gel images and SAXS curves, it could be beneficial to also include representative fluorescence anisotropy titrations for the binding data in the tables.

2.     Line 388, do the authors think that the mobilization of the PBS/SL1 region would be a considered a chaperone activity by Phospho-LysRS?

3.     Fig. 5C. I think it would be helpful to add another orthogonal view, to more clearly visualize the progressive evolution of the molecular envelope with the DNA extension.

4.     I believe that it would help the readers if the authors include a cartoon scheme of the now revised structural model, with a dimeric P-LysRS bound to a dimeric HIV RNA UTR.  

Author Response

Response to Reviews

RE: MS # viruses-1775293

Reviewer 1

Cantara et al., presents a comprehensive study of the interactions between the host Lysyl-tRNA synthetase (LysRS) and a HIV-1 5’-UTR region, including the primer-binding site (PBS), PBS-proximal TLE (tRNA-like element), and packaging signal stem loop 1 (SL1). Combining crosslinking, chemical probing, phosphomimetics, SAXS, fluorescence anisotropy, and structure modeling, the authors discovered that LysRS phosphorylation led to a more open conformation of the UTR when bound to LysRS, and enhanced flexibilities of the PBS and SL1 regions, which may facilitate the annealing of the tRNA primer. Based on crosslinking results, the authors propose a revised structural model, where an extended phospho-LysRS dimer binds bivalently to a HIV-1 UTR dimer via the distant TLE and SL1 interfaces. Since phosphorylated LysRS is the functional form that is packaged in virions, this study provides important new insights into the mechanisms of retroviral replication. The manuscript is expertly written, clearly illustrated, technically rigorous, and is both thorough and thoughtful. I enthusiastically support the publication of this outstanding study, and have but a few minor, largely cosmetic suggestions below.

 Specific comments:

  1. Since the authors already include raw gel images and SAXS curves, it could be beneficial to also include representative fluorescence anisotropy titrations for the binding data in the tables.
    • We have added our fluorescence anisotropy data to the supplementary material as new Figures S3 and S7.
  1. Line 388, do the authors think that the mobilization of the PBS/SL1 region would be a considered a chaperone activity by Phospho-LysRS?
    • Generally, we think of chaperones as functioning to destabilize misfolded RNA intermediates or to promote folding into the native state. It is typical for chaperones to accomplish this by manipulating the dynamics of the target RNA before rapid dissociation. However, we feel that the evidence we present herein does not rise to the level of chaperone activity because we don’t see the PBS/TLE as being folded into a non-native intermediate that is relieved by phosphor-LysRS binding. Also, due to its speculative nature, the statement on lines 387-389 of the original submission that read “The XL-SHAPE data suggest that binding of phosphorylated LysRS may facilitate destabilization of the 5′UTR prior to tRNA primer annealing and reverse transcription initiation.” was removed from the results section.
  1. Fig. 5C. I think it would be helpful to add another orthogonal view, to more clearly visualize the progressive evolution of the molecular envelope with the DNA extension.
    • We agree but felt that this would be more appropriate in supplementary material so we have added this as new Figure S9.
  1. I believe that it would help the readers if the authors include a cartoon scheme of the now revised structural model, with a dimeric P-LysRS bound to a dimeric HIV RNA UTR.
    • A cartoon scheme has been added as Figure 6. We have also added the following text to the last paragraph of the discussion to properly explain the new figure: “Based on these results, a new working model is proposed (Figure 6). In this model, HIV-1 infection triggers phosphorylation of LysRS at S207, resulting in LysRS release from the MSC in a conformation that is competent for tRNA binding but not for aminoacylation [11,15] (Figure 6, Step 1). The catalytically inactive dimeric pS207-LysRS/tRNA complex is then recruited to sites of viral assembly via interactions with HIV-1 Gag [5,48]. The 5’UTR can adopt various monomeric and dimeric conformations, including a “kissing loop” dimer [49-54] (Figure 6, Step 2). The anticodon-like TLE of an HIV-1 gRNA dimer then competes for binding to pS207-LysRS resulting in an increase in dynamics of the gRNA in this region and release of the tRNA primer proximal to the PBS (Figure 6, Step 3). Crosslinking and dynamic changes are also observed in the SL1 hairpin, but whether a second LysRS dimer binds directly to this region as shown here remains to be confirmed. The tRNA primer is subsequently annealed to the PBS via chaperone activity of the nucleocapsid domain of the HIV-1 Gag polyprotein [55,56] (Figure 6, Step 4). After viral budding and maturation, reverse transcriptase specifically binds the tRNA:gRNA duplex (Figure 6, Step 5) and initiates synthesis of minus-strand strong stop proviral DNA from the 3′ end of the uncharged tRNA primer (Figure 6, Step 6). During initial extension of the primer (3-11 nt), the conformation of the TLE changes and its tRNA mimicry is lost, triggering release of pS207-LysRS from the gRNA. Based on this model, we propose that pS207-LysRS initially selects and facilitates tRNA annealing on the kissing loop dimer subpopulation of gRNA conformers. Annealing of the tRNA primer further stabilizes the kissing dimer and mature nucleocapsid protein subsequently promotes formation of a more stable extended-dimer conformation [49].”

Reviewer 2

The study of Cantara et al. builds on previous reports from the Musier-Forsyth laboratory showing that Lysyl-tRNA synthetase (LysRS) is selectively packaged into HIV-1. In HIV-1 infected cells, LysRS is phosphorylated on S207, resulting in its release from the multi-aminoacyl-tRNA synthetase complex (MSC) and partial re-localization to the nucleus. Pharmacological inhibition of S207 phosphorylation reduces viral infectivity. A phosphomimetic variant of LysRS (S207D), lacking aminoacylation activity, is also localized to the nucleus and packaged into HIV-1 virions. In this new study, the authors address how S207 phosphorylation affects interaction with the gRNA. The authors conduct an in vitro comparison of WT and S207D-LysRS interactions with the HIV-1 gRNA. Results suggest that S207D-LysRS preferentially binds dimeric gRNA in an open conformation distinct from non-phosphorylated protein. Results also show that S207D-LysRS destabilizes the PBS region and SL1.

The manuscript is well written, and the data are clearly structures and presented. Results are exciting and meaningful for RNA structure-function experts, especially those working in HIV-1. However, some sections may need clarification for virologists with little expertise in RNA structure-function.

Comments.

1- The pNL4.3 5’UTR initiates at nt 1, finishing at position 336 (Gag initiation codon). However, the authors use a segment that leaves out the TAR and PolyA loop (1-104) and extends within the Gag coding region (337-345), nts 106-345. It might be straightforward for an expert to understand that authors wish to stabilize the RNA structure that determines gRNA encapsidation by favoring interactions between nts 105-115 and 334-345 (Science. 2011;334(6053):242-5. doi: 10.1126/science.1210460), but for a nonexpert, this might not necessarily be evident. Thus, this should be clearly stated in the text.

  • We have clarified this point by adding the following text at lines 318-321 “The first nine nucleotides of the gag coding region are included because they contribute to the structure and stability of the packaging signal [38], but the first 104-nt (TAR and polyA hairpins) are excluded because they do not appreciably contribute to LysRS binding [17].” We noted the rationale for inclusion of the mutations in U5 and AUG as being for stabilization and to facilitate in vitro transcription using T7 RNA polymerase (Methods section lines 150-153 and Figure 1 legend).

2- The numbering and color pattern in Fig. 1B should be verified and referenced. For example, the figure legend indicates “Psi domain (blue, nt 106 to 345)”; however, blue covers only until nt 332. Nts 333-345 are in black. A reference should be provided to support the structure presented in Fig. 1B as it does not fully recapitulate interactions reported by others. For example, the interaction between U117-A332 is not in (Science. 2011;334(6053):242-5. doi: 10.1126/science.1210460; Proc Natl Acad Sci U S A. 2014;111(9):3395-400. doi: 10.1073/pnas.1319658111).

  • The numbers in the text of the Figure 1B figure legend have been corrected and the following text was added to reference the secondary structure presented: “The secondary structure shown is based on Ref 38.”

3- In the case of the PBS/TLE105 what evidence do authors have to conclude that this region alone conserves its structural features? Shouldn’t the PBS/TLE105 be flexible in structure as it harbors a single-stranded region? (Proc Natl Acad Sci U S A. 2014;111(9):3395-400. doi: 10.1073/pnas.1319658111).

  • In the referenced paper (PNAS. 2014;111(9):3395-400), we used small-angle X-ray scattering (SAXS) to produce an all-atom model of this exact PBS/TLE105 RNA construct. Prior to that study, the secondary structure of that construct had not been unambiguously determined. We therefore screened the most commonly predicted secondary structures (see text description in the second to last paragraph of page 3397 and Figure S6 of [PNAS. 2014;111(9):3395-400]). In that paper, we determined that the secondary structure we depict in the current manuscript best recapitulated that experimental SAXS data.

4- Speculation should be left for the discussion section. Results show that LysRS destabilizes the structure of the targeted RNA. However, if this occurrence has any relationship with tRNA priming or reverse transcription is highly speculative in the context of the shown data (lines 387-388). The same is valid for lines 455-457.

  • These speculative sentences have been removed from the text.

Reviewer 2 Report

The study of Cantara et al. builds on previous reports from the Musier-Forsyth laboratory showing that  Lysyl-tRNA synthetase (LysRS) is selectively packaged into HIV-1. In HIV-1 infected cells, LysRS is phosphorylated on S207, resulting in its release from the multi-aminoacyl-tRNA synthetase complex (MSC) and partial re-localization to the nucleus. Pharmacological inhibition of S207 phosphorylation reduces viral infectivity. A phosphomimetic variant of LysRS (S207D), lacking aminoacylation activity, is also localized to the nucleus and packaged into HIV-1 virions. In this new study, the authors address how S207 phosphorylation affects interaction with the gRNA. The authors conduct an in vitro comparison of WT and S207D-LysRS interactions with the HIV-1 gRNA. Results suggest that S207D-LysRS preferentially binds dimeric gRNA in an open conformation distinct from non-phosphorylated protein. Results also show that  S207D-LysRS destabilizes the PBS region and SL1.

The manuscript is well written, and the data are clearly structures and presented. Results are exciting and meaningful for RNA structure-function experts, especially those working in HIV-1. However, some sections may need clarification for virologists with little expertise in RNA structure-function.

Comments.

1- The pNL4.3 5’UTR initiates at nt 1, finishing at position 336 (Gag initiation codon). However, the authors use a segment that leaves out the TAR and PolyA loop (1-104) and extends within the Gag coding región (337-345), nts 106-345. It might be straightforward for an expert to understand that authors wish to stabilize the RNA structure that determines gRNA encapsidation by favoring interactions between nts 105-115 and 334-345 (Science. 2011;334(6053):242-5. doi: 10.1126/science.1210460), but for a nonexpert, this might not necessarily be evident. Thus, this should be clearly stated in the text.

2- The numbering and color pattern in Fig. 1B should be verified and referenced. For example, the figure legend indicates “Psi domain (blue, nt 106 to 345)”; however, blue covers only until nt 332. Nts 333-345 are in black. A reference should be provided to support the structure presented in Fig. 1B as it does not fully recapitulate interactions reported by others. For example, the interaction between U117-A332 is not in (Science. 2011;334(6053):242-5. doi: 10.1126/science.1210460; Proc Natl Acad Sci U S A. 2014;111(9):3395-400. doi: 10.1073/pnas.1319658111).

3- In the case of the PBS/TLE105 what evidence do authors have to conclude that this region alone conserves its structural features? Shouldn’t the PBS/TLE105 be flexible in structure as it harbors a single-stranded region? (Proc Natl Acad Sci U S A. 2014;111(9):3395-400. doi: 10.1073/pnas.1319658111).

4- Speculation should be left for the discussion section. Results show that LysRS destabilizes the structure of the targeted RNA. However, if this occurrence has any relationship with tRNA priming or reverse transcription is highly speculative in the context of the shown data (lines 387-388). The same is valid for lines 455-457.

Author Response

(The authors gave the same response as above.)
